# Systematic Identification and Analysis of *OSC* Gene Family of *Rosa rugosa* Thunb

**DOI:** 10.3390/ijms232213884

**Published:** 2022-11-11

**Authors:** Jianwen Wang, Pengqing Wang, Mengmeng Xu, Yudie Chen, Liguo Feng

**Affiliations:** College of Horticulture and Landscape Architecture, Yangzhou University, Yangzhou 225009, China

**Keywords:** *Rosa rugosa* Thunb, oxidosqualene cyclase, biosynthesis, triterpenes

## Abstract

The oxidosqualene cyclase family of *Rosa rugosa* (*RrOSC*) provides a starting point for the triterpenoid pathway, which contributes to the medicinal value of the extraction of tissues of *Rosa rugosa*. However, the structure and function of key *RrOSCs* of active triterpenoids remain ambiguous. In this study, a total of 18 *RrOSC* members with conservative gene structures and motifs were identified based on the genome of *Rosa rugosa*. The *RrOSCs* were located on three chromosomes including two gene clusters that derived from gene replication. The phylogenetic relationship divided *RrOSCs* into six groups, and the *RrOSCs* of GI and GIV that were represented by lupeol or α-amyrin were identified as likely to include candidate genes for producing active triterpenoids. Considering the high expression or specific-tissue expression of the candidates, *RrOSC1*, *RrOSC10*, *RrOSC12*, and *RrOSC18* were considered the key genes. *RrOSC12* was identified in vitro as lupeol synthase. The results provided fundamental information and candidate genes for further illustration of the triterpenoid pathway involved in the pharmacological activities of *Rosa rugosa*.

## 1. Introduction

Sterols and triterpenes are important components of plant metabolites, and nearly 200 different skeletons of these compounds have been identified [1]. In addition to components of cell membranes and roles in plant growth and development, triterpenes perform key roles in commercial applications by biotic or abiotic function, such as preservatives, flavor modifiers, and cholesterol-lowering agents. For example, the sweetness of licorice roots is attributable to the presence of the triterpenoid saponin glycyrrhizin. Evidence of the health benefits of saponins is increasing, and their expanding application in food, cosmetics, and pharmaceutical industries is attracting commercial attention [2]. In the aspect of biochemical reaction, 2,3-Oxidosqualene is the common precursor of all the sterols and triterpenoids. The cyclization processes of 2,3-Oxidosqualene including protonation, cyclization, rearrangement, and deprotonation produce the diversiform skeletons of sterols and triterpenes [3,4,5]. During the cyclization processes, “chair-chair-chair (C-C-C)” and “chair-boat-chair (C-B-C)” configurations are present prior to cyclization into diverse triterpenes. Sterols are mainly formed through the “C-B-C” configuration, and triterpenoids are mainly formed through the “C-C-C” configuration [1].

Oxidosqualene cyclases (OSCs) are the pivotal enzyme in promoting the diversification of triterpenes, which can catalyze cyclization processes. Phillips et al., (2006) [6] divided plant OSCs into two groups based on the nature of their presumed catalytic intermediates, that is, protosteryl and dammarenyl cations, which possess distinct stereochemistry and ring configurations [6,7]. The protosteryl cation adopts the C-B-C configuration and is an intermediate leading to the formation of cycloartenol, lanosterol, parkeol, and cucurbitadienol tetracyclic triterpene skeletons (6-6-6-5). Isoarborinol is an unusual pentacyclic triterpene (6-6-6-6-5) derived from the additional D-ring expansion of the protosteryl cation based on the C-B-C configuration. Most pentacyclic triterpene skeletons, however, are derived from the dammarenyl cation by D-ring expansion to form lupeol or further E-ring expansion to form β-amyrin via the C-C-C conformation [1]. More than 150 *OSC* genes have been identified in plants and their catalytic function were mostly identified by heterologous expression in yeast. The *Arabidopsis thaliana* genome contains 13 *OSC* genes, including *cycloartenol synthase* (*AtCAS1*), *lanosterol synthase* (*AtLSS1*), and *β-amyrin synthase* (*LUP4*), and several *OSCs* with mixed functions [8,9,10,11,12]. The *amyrin synthase* of *Eriobotrya japonica* (*EjAS*) can catalyze the conversion of 2,3-oxidosqualene into α- and β-amyrin at a ratio of 17:3 [13]. In *OSCs* of *Malus×domestica* (*MdOSCs*), *MdOSC1* and *MdOSC3* can produce α-amyrin as the main product, with β-amyrin and lupeol as the minor products. *MdOSC4* is characterized as germanicol synthase, and *MdOSC5* produces lupeol as a major product [13,14]. Further, the potential of plant *OSCs* has been exploited in engineered yeast. *MdOSC1* introduced *Saccharomyces cerevisiae* achieve 5.8 times the maximum α-amyrin production [14]. 

*Rosa rugosa* is a shrub of Rosaceae and native to North China, Northeast China, Japan, Korea, and Russia, and has a long history of application as a medicinal plant. Its petals and roots have been widely used to staunch bleeding [15], reduce blood lipids, treat diabetes, and as a cure for cancer or cardiovascular diseases [16,17]. In the *R. rugosa* root extract, the active triterpenoids Kaji ichigoside F1, rosamultin, euscaphic acid, and tormentic acid showed anti-injury and anti-inflammatory effects on animal models; among these compounds, rosamultin had a strong inhibitory effect on human immunodeficiency virus-1 protease [18,19]. These reports point to the key role that terpenoids play in the pharmacological activities of *R. rugosa*. However, the *OSC* genes of *R. rugosa* (*RrOSCs*) involved in the synthesis of pharmacological terpenoids (e.g., rosamultin) haven’t been reported. This study was aimed at discovering the potential key *RrOSCs* in the active terpenoids synthesis pathway of *R. rugosa* by systematic analysis of the *RrOSC* family.

## 2. Results

### 2.1. Identification and Phylogenetic Analysis of RrOSCs

A total of 18 *RrOSC* genes were identified, and the predicted proteins ranged from 540 to 770 amino acids in length and from 62.283 to 85.712 kDa in molecular weight (Appendix A). All *OSCs* (except *RrOSC4*, which is extremely short and could belong to a pseudogene or unknown coding gene) were divided into 6 Groups (GI–GVI), which were named by common catalytic function based on the phylogenetic branches (Figure 1). *OSCs* in GI would produce mixed products and the main components of the mixed products were different; *OSCs* of GIII including *RrOSC9* and *RrOSC13* would be cycloartenol synthases. *OSCs* of GIV including *RrOSC12* would be lupeol synthases. *OSCs* of GV including *RrOSC14*, *RrOSC 15*, and *RrOSC 16* would be lanosterol synthases. No *RrOSCs* belonged to GII (β-amyrin synthases) or GVI (specific to monocotyledons).

### 2.2. The Conserved Motifs and Gene Structures of RrOSCs

The exon number of *RrOSCs* was in the range of 14–21. The SQHop_cyclase_C and/or SQHop_cyclase_N domains located by constant exon number (18–20 exons) indicated gene structures of *RrOSCs* were conserved (Figure 2b). Analysis of motifs (Figure 2a) showed that the domains of most *RrOSCs* composed of motifs 3-8-5-10-2 (corresponding to SQHop_cyclase_C) and motifs 6-7-4-1 (corresponding to SQHop_cyclase_N) were highly aligned. Interestingly, Group I members lack one motif5 in the N-terminal compared while other *RrOSCs* comprised of two motif5. And motif6 and/or motif7 were absent in SQHop_cyclase_N of Group I members *RrOSC1* and *RrOSC3*.

### 2.3. Clusters and Gene Duplications of RrOSCs

The *RrOSCs* are distributed on three (Chr2, Chr4, and Chr7) of all seven chromosomes (Figure 3a). Most members are located on the gene clusters of Chr2 and Chr4 while only *RrOSC18* is distributed on the Chr7. According to the gene duplication type (Appendix A), *RrOSC2/3/5/8/9/14/15* belonged to the tandem duplication type and *RrOSC6/7/10/13* belonged to the proximal duplication type. According to the gene pairs of synteny analysis (Appendix A), two tandemly duplicate gene pairs *RrOSC2-RrOSC3*, *RrOSC8-RrOSC9* (Figure 3b), and proximal duplicated *RrOSC6/7/10* contributed to the origin of the gene cluster on Chr2. *RrOSC14-RrOSC15* pair (Figure 3b) and proximal duplicated *RrOSC13* explained the origin of the gene cluster on Chr4. The segmentally duplicate events (Figure 3b) were not detected in the *RrOSCs*. 

### 2.4. Expression Analysis of RrOSCs

To detect the candidate *RrOSC* genes of the synthetic pathways of *R. rugosa* active triterpenoids, we investigated the temporal and spatial expression patterns of *RrOSCs* in roots, flowers, and lateral branches (Figure 4). *RrOSC*2, *RrOSC*3, and *RrOSC*5 of GI and *RrOSC*14, *RrOSC*15, and *RrOSC*16 of G5 expressed in a low abundance (TPM < 1) were not discussed further. *RrOSC9*, *RrOSC10*, and *RrOSC13* are expressed in all tissues and all stages in a middle or high abundance (minimum TPM > 10), indicating their constitutive expression pattern.

The highly-expressed *RrOSC6*, *RrOSC10*, and *RrOSC13* in the lateral branches of *R. rugosa* showed two expression patterns. The expression of *RrOSC6* and *RrOSC13* increased from the first lateral branch to the third lateral branch. The expression of *RrOSC10* was the highest, decreased from the primary lateral branch to the secondary lateral branch, and increased from the secondary lateral branch to the tertiary lateral branch. The three genes all showed the highest abundance in the third lateral branch, and only *RrOSC6* specific expressed in branches.

In roots (Figure 4, TIR or OR), the expression levels of *RrOSC10*, *RrOSC12*, *RrOSC13*, and *RrOSC18* were high (FPKM > 50), and *RrOSC12* was strongly expressed in TIR but almost not at all in OR. In flowers, the expression levels of *RrOSC9*, *RrOSC10*, and *RrOSC13* were higher. The expression levels of *RrOSC9* and *RrOSC13* were high in the early stage, decreased at the end of the flowering stage, and reached a maximum at S2 and S3. The expression of *RrOSC10* was the highest, and it increased significantly at the end of the flowering stage and reached the maximum at S7. In the lateral branches and petals, more than half of the genes were not expressed or had low expression.

### 2.5. Functional Characterization of RrOSC12

To elucidate the function of *RrOSC12*, we detected its products by expression in yeast. Authentic lupeol showed a dominant peak at 17.8 min (Figure 5b). A peak with the corresponding retention time was observed in the total ion chromatogram of *RrOSC12* expressed yeast, and mass spectra confirmed that it was that of lupeol (Figure 5c). These results demonstrate that *RrOSC12* catalyzes 2,3-oxidosqualene to yield lupeol. 

## 3. Discussion

The *OSC* gene number (18 members) of *R. rugosa* was more than the 14 members found in Arabidopsis and 11 members in rice. A comparison of gene structure and motifs showed that the gene structure difference of *RrOSCs* in the same subgroup didn’t affect their highly-conserved domains including consistent motif structure. Additionally, many *RrOSCs* in the same subgroup shared more similar expression patterns, such as two different subgroups of GI: *RrOSC7*, *RrOSC11*, and *RrOSC17* all highly expressed in flowers or *RrOSC6*, *RrOSC10*, and *RrOSC18* all highly expressed in roots (or branches).

The OSCs in the same subgroup in GI could produce mix-products with accordant main components. *RrOSC1*-*RrOSC6*, *RrOSC10*, and *RrOSC18* were clustered with *MdOSC3*, *EjAS*, and *MdOSC1* of Rosaceae plants as a subgroup. *MdOSC1* yields α-amyrin, β-amyrin, and lupeol at a ratio of 85:13:2 [13], *EjAS* produces α-amyrin and β-amyrin at a ratio of 17:3, and *MdOSC3* manufactures α-amyrin, β-amyrin, and lupeol at a ratio of 85:14:1 [20]. This subgroup was most likely a cluster of α-amyrin synthases. In addition, the tandem replication pair *RrOSC2-RrOSC3* and proximal duplication genes *RrOSC6*, and *RrOSC10* indicated that gene duplications contributed to the functional redundancy of the seven α-amyrin synthase candidates. *RrOSC7*, *RrOSC8*, *RrOSC11*, and *RrOSC17* combined with *MdOSC4* and *MdOSC5* were clustered into another subgroup. The main product of *MdOSC4* is germanicol, that of *MdOSC5* is lupeol, and both contain the by-product β-amyrin. Thus, the four candidates are unlikely to be α-amyrin synthases. According to the structural comparison, the rosamultin of *R. rugosa* should be formed from α-amyrin through cytochrome P450 oxidation to introduce functional groups and finally glycosylated by glycosyltransferase. Except for the four low abundance genes, above, *RrOSCs*, *RrOSC1*, *RrOSC10*, and *RrOSC18* were considered as the candidate genes of the rosamultin synthesis pathway. 

Lupeol is an important pentacyclic triterpenoid. The applications of lupeol derivatives in the field of medicine are extensively studied because of their pharmacological activities, including antinephrotoxicity, antihepatotoxicity, and anticancer, anti-inflammatory, anti-heart disease, antiarthritis, and antidiabetes properties [21]. Lupeol was also found to be involved in the formation of epicuticular wax crystals on the stem and hypocotyl surfaces [22]. Only *OSC* of *R. rugosa* (*RrOSC12*) belongs to GIV, whose members would catalytic substrate to produce lupeol. The yeast expression proved its catalytic activity of lupeol. Though the lupeol synthase activity in vivo of *RrOSC12* needs to be studied further, its remarkably high mRNA abundance in TR and the lupenone and betulinic acid (downstream products of lupeol) accumulation in TR (unpublished metabolome data) indicated that lupeol contributes partly to the medicinal value of *R. rugosa* roots. In addition, the chromosome terminal locations (near telomere) of *RrOSC12* provide insight into DNA methylation, which could be the entry point for further exploring the pathway gene cluster to jointly modify the substrate (i.e., the gene cluster of *RrOSC12* and corresponding downstream P450 genes).

The enormous metabolic engineering potential of *OSCs* is increasingly apparent, e.g., *MdOSC1* [14,23], *enzymes cycloartenol synthase gene* of *Panax ginseng* and *β-amyrin synthase genes* of *Glycyrrhiza glabra*, and *P. ginseng* [24,25]. Based on the genetic transformation method in hairy roots of *R. rugosa* that has been established by our own lab, we would like to overexpress *RrOSC12* in *R. rugosa* hairy roots to increase lupeol production. In addition, the two α-amyrin synthase candidates *RrOSC10/18*, which were highly expressed in roots, should be tested further for increasing the potential of α-amyrin and that of the downstream product rosamultin.

## 4. Materials and Methods

### 4.1. Plant Materials

The three-year-old cutting seedings of *R. rugosa* “*Zi Zhi*” were planted in germplasm in the resource nursery of Yangzhou University (32.391° N, 119.419° E), Yangzhou, China. Samples including roots (OR), primary lateral branch (PB), secondary lateral branch (SB), tertiary lateral branch (TB), flowers of seven stages (S1–S7), and roots (TR) of tissue culture seedlings (two-month-old micro-cutting seedings) were collected and frozen using liquid nitrogen, and then stored at −80 °C for RNA extraction (Figure 6).

### 4.2. Identification of RrOSCs

*RrOSCs* were predicted by HMMER 3.0 from the proteome of *R. rugosa “Zi Zhi”* (unpublished) using the hidden Markov model of the squalene-hopene cyclase N-terminal domain (PF13249) and squalene-hopene cyclase C-terminal domain (PF13243) [26]. The candidate genes with both domains were confirmed by Pfam (http://pfam.xfam.org/, accessed on 1 September 2022) and the Conserved Domain Database [27,28]. The dubious *OSC* genes were checked by manual inspection. The isoelectric point and molecular weight of RrOSC proteins were obtained by ProtParam tool of Extasy [29].

### 4.3. Phylogenetic Analyses

RrOSC proteins and 78 OSC proteins of different plant species were aligned using the MAFFT algorithm (https://mafft.cbrc.jp/alignment/server/index.html, accessed on 1 September 2022) [30]. The phylogenetic dendrogram was constructed by the neighbor-joining method of MEGA-X [31], with 1000 bootstrap replications. In addition, the p-distance model of the substitution type, pairwise deletion of Gaps/Missing data treatment, and uniform rates of rates among sites were selected for phylogenetic analysis. The 78 *OSC* genes of *Arabidopsis thaliana*, *Lotus japonicus*, *Panax ginseng*, *Abies magnifica*, *Adiantum capillus-veneris*, *Avena strigose*, *Betula platyphylla*, *Costus speciosus*, *Cucurbita pepo*, *Dioscorea zingiberensis*, *Glycyrrhiza glabra*, *Kalanchoe daigremontiana*, *Kandelia candel*, *Luffa cylindrica*, *Oryza sativa*, *Pisum sativum*, *Polypodiodes niponica*, *Rhizophora stylosa*, *Ricinus communis*, *Artemisia annua*, *Aster sedifolius*, *Bruguiera gymnorrhiza*, *Euphorbia tirucalli*, *Medicago truncatula*, *Nigella sativa*, *Polygala tenuifolia*, *Solanum lycopersicum*, *Vaccaria hispanica*, *Olea europaea*, *Taraxacum officinale*, *Aster tataricus*, *Stevia rebaudiana*, *Malus pumila*, and *Eriobotrya japonica* are shown in Appendix A.

### 4.4. Gene Structures and Conserved Motifs Analysis

Conserved motifs of RrOSC proteins were predicted by the MEME suite (http://meme-suite.org/, accessed on 1 September 2022) with the default parameters and a maximum motif number of 8 [32]. The conserved motifs and gene structures with domain location of *RrOSCs* were illustrated by the gene structure view tool of TBtools toolkit 1.086 [33] based on the genomic DNA and mRNA sequences of the *RrOSCs* (Appendix A).

### 4.5. Chromosome Location and Synteny Analysis

According to the GFF3 file of the reference genome, *RrOSC* gene location was illustrated by the Gene location visualize function of the GTF/GFF tool of TBtools toolkit 1.086 [33].

Inter-species synteny analysis was conducted by reciprocal BLASTP search for potential homologous gene pairs (E < 10 − 5, top five matches) of the whole genome, and the syntenic regions were predicted by the MCScanX tool of TBtools toolkit 1.086 [34]. The gene duplication events were predicted according to syntenic regions by the ‘duplicate_gene_classifier’ program of MCScanX [34]. 

### 4.6. Expression Analysis and Functional Characterization

The RNA of the above 12 tissues (R, TR, PB, SB, TB, and S1–S7) were extracted by the MiniBEST Universal RNA Extraction Kit (TaKaRa, Kusatsu, Japan) for transcriptome libraries building and sequencing on the Illumina NovaSeq 6000 platform. The raw reads could be accessed in the SRA of NCBI (PRJNA725330) and BIG Sub of China National Center for Bioinformation (PRJCA012932). After low-quality reads filtering, clean reads were used for analysis. Based on clean reads, read counts of each gene were calculated by mapping to the genome of *R. rugosa* “*Zi Zhi*” using HISAT 2.2.1 (http://daehwankimlab.github.io/hisat2/, accessed on 1 September 2022). The sequencing quality was examined by Q30 value, percentage of clean reads, and alignment coverage (Appendix A). Fragments per kilobase of exon model per million reads mapped (FPKM) of *RrOSCs* were calculated by normalization of read counts using a self-built R script.

*RrOSC12* was cloned by the polymerase chain reaction (PCR). The coding sequence of *RrOSC12* was sub-cloned to pESC-leu and transformed to yeast strain AM94. Culture extracts of the *RrOSC12* expressed yeast were derivatized with trimethylsilylating agents and submitted to gas chromatography–mass spectrometry analysis.

## Figures and Tables

**Figure 1 ijms-23-13884-f001:**
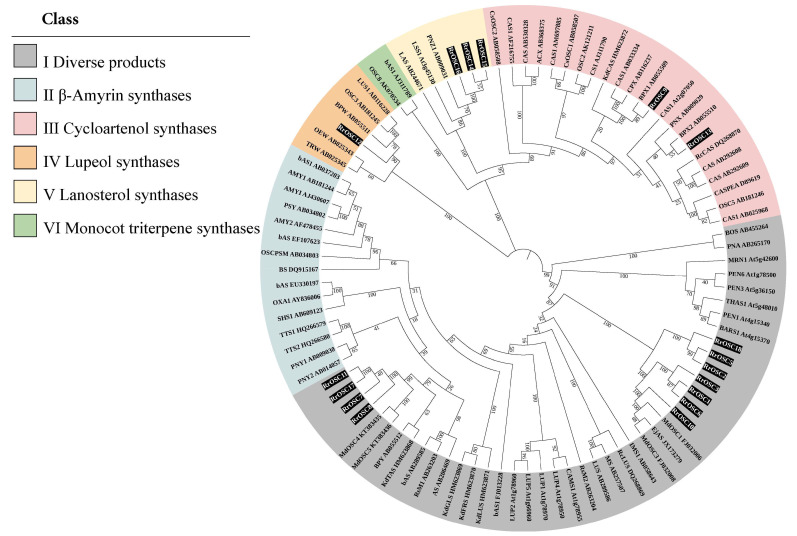
Phylogenetic analysis of 95 *oxidosqualene cyclases* (*OSCs*) of different plant species. The phylogenetic dendrogram was generated using the neighbor-joining method with 1000 bootstrap replicates (numbers in branches). Groups were distinguished with different colors and *OSCs* of *Rosa rugosa* (*RrOSCs*) were highlighted. The information on plant species and corresponding *OSCs* are listed in Appendix A.

**Figure 2 ijms-23-13884-f002:**
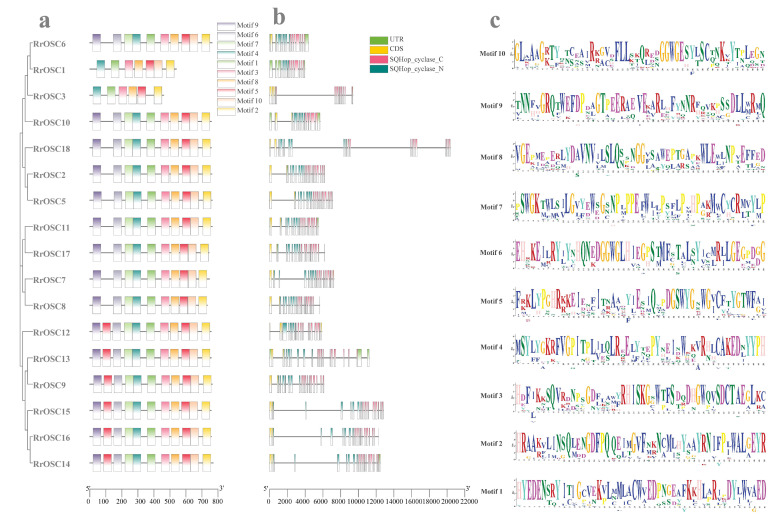
Motifs (**a**), gene structures (**b**), and motif sequences (**c**) of *RrOSCs*. The coloring boxes represent motifs whose locations were labeled by the scale plate of amino acid residue (**a**). Exons (rectangles) separated by introns (lines) were colored yellow (coding sequence) and green (untranslated region), and the locations were labeled by the nucleotide scale plate. Domains were colored purple (SQHop_cyclase_C) and blue (SQHop_cyclase_N) (**b**). The top 10 significantly enriched motifs are listed as sequence logos (**c**).

**Figure 3 ijms-23-13884-f003:**
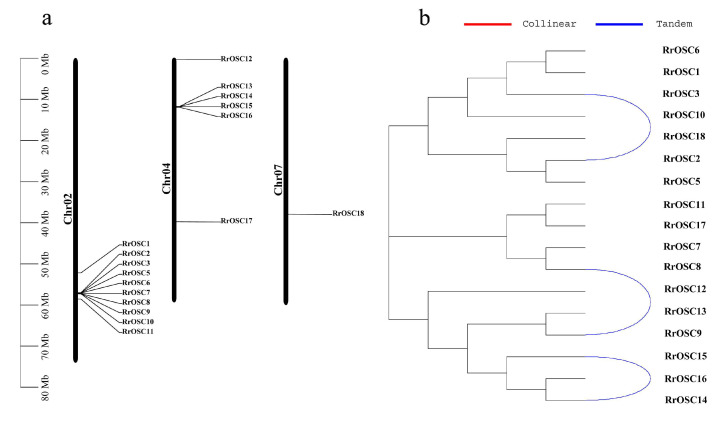
Chromosomal localization (**a**) and tandem duplication (**b**) analysis of *RrOSCs*. Blue lines indicated the duplication gene pairs according to the synteny analysis.

**Figure 4 ijms-23-13884-f004:**
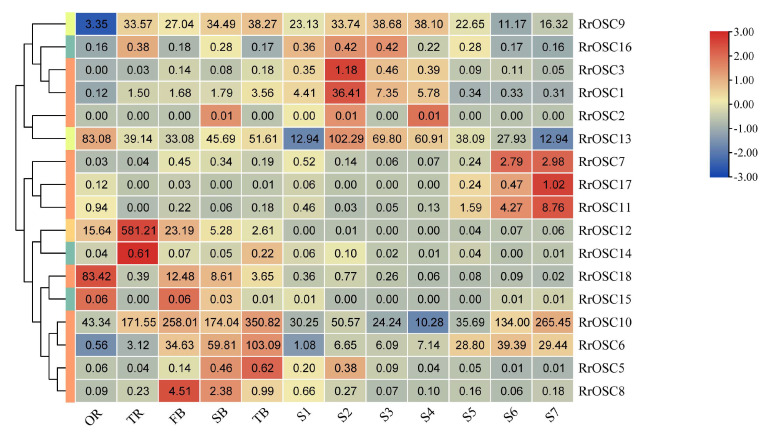
Digital gene expression profiles in the roots, lateral branches, and flowers of *R. rugosa* “Zi Zhi”. A heat map was generated based on the normalized Log2 FPKM represented by the blue-yellow gradation. The numbers in the heat map were FPKM from RNA-seq data. The OR and TR columns represent the roots of open-air and tissue culture seedlings, respectively The columns of FB, SB, and TB represent the primary, secondary, and tertiary lateral branches, respectively. The columns of S1–S7 represent the seven flower stages, respectively.

**Figure 5 ijms-23-13884-f005:**
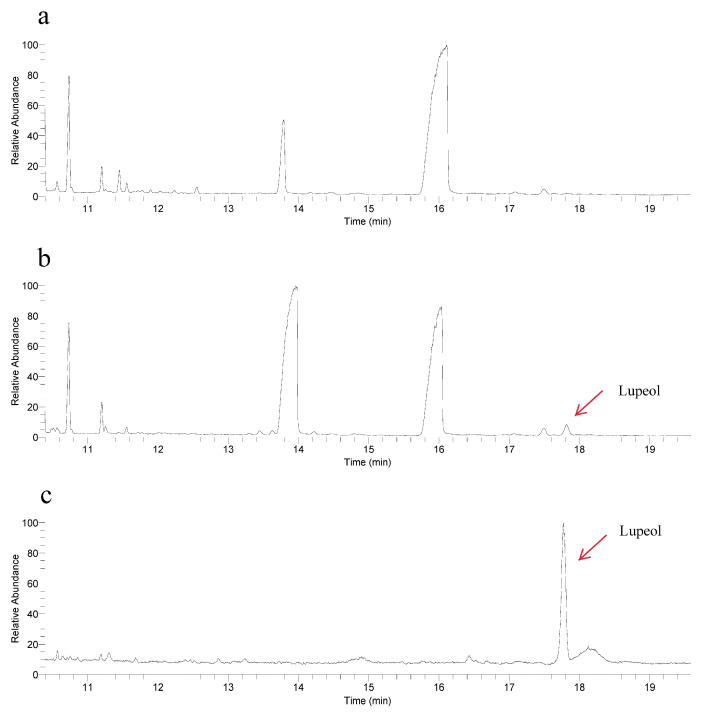
Gas chromatography–mass spectrometry analysis of yeast extracts overexpressing the *R. rugosa* lupeol synthase gene. Chromatogram for the yeast with empty vector (**a**), transformed yeast with the *RrOSC12* gene (**b**), and the authentic lupeol standard (**c**). Mass spectra in a retention time of 17.8 min for lupeol in the transformed yeast (**d**) and the authentic lupeol standard (**e**).

**Figure 6 ijms-23-13884-f006:**
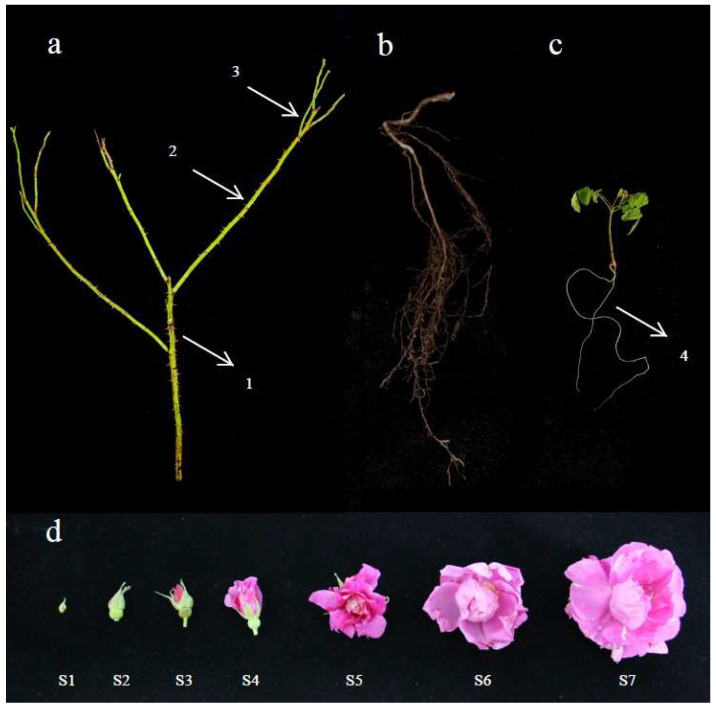
The lateral branches (**a**), primary lateral branch (1), secondary lateral branch (2), tertiary lateral branch (3). Roots (**b**) and roots of tissue culture seedlings (**c**) (4). Petals at different development stages (**d**), S1: Large bud stage, S2: Reddish stage, S3: Flowering initiation stage, S4: Flower bud half-opening stage, S5: Initial opening stage, S6: Semi-opening stage, S7: Full opening stage.

## Data Availability

Not applicable.

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
