# Peer review of "Systematic Identification and Analysis of OSC Gene Family of Rosa rugosa Thunb"

_ijms, 2022, doi:10.3390/ijms232213884_

Round 1

Reviewer 1 Report

The manuscript " Systematic Identification and Analysis of OSC Gene Family of Rosa rugosa Thunb."  in my opinion, presents interesting and valuable data. It fits in well with the scope of the journal. However, the manuscript requires major revisions and supplementing important information. Particularly the Materials and Methods are not sufficiently described, and this section needs to be completed.

1.     Figures 1. and 2. are small and unreadable. Figure 2. should be enlarged and possibly divided. Instead, you can remove Figure 6, which in my opinion, is not necessary (or move it to the supplement)

2.     The authors several times n the article indicated they identified 18 OSC genes, but the RrOSC4 gene is missing in table S1 and in all analyses. All information about this gene is missing except that it is short and possibly a pseudogene ( line 72). Size and chromosomal location have not been estimated. How was this gene identified, and why was it not included in the analysis? Was it observed in the transcriptome?

3.     What did Gene IDs mean in Table S1? Are the sequences of RsOSC genes accessible in any databases? The sequences of all new RrOSC genes should be submitted to the database, and the accession numbers should be enclosed in the article.

4.     Complete table S1 with additional information: length of the nucleotide sequence, number of exons, and location on the chromosome.

5.     Line 211. Identification of RrOSCs   description is insufficient. Based on the text, I can't understand how the authors obtained the RrOSCAs amino acid sequences. No information about reference sequences was used to predict the OSCs sequences.

6.     No information about the proteome of R. rugosa "Zi Zhi." Is it accessed in any database? Where is the proteome come from? What tissue was this transcriptome prepared?

7.     The authors poorly describe the process of gene identification and their location in the genome. For example, what bioinformatics tools were used for genome localization? What was the reference genome?

8.     Line 216. The subcellular localization –  where is the result of the subcellular localization analysis described in the article?

9.     Phylogenetic analyses - lack basic information regarding phylogenetic analysis:  alignment methods and program settings. What substitution model was used in the phylogenetic tree construction, and how was this model established? A table containing the percentage of gene similarity at the AA level would be helpful. It could be added to the supplement.

10.  Line 231 - Information about NGS sequencing results should be provided: sequencing quality - Q30 value, percentage of rejected reads after quality trimming, and alignment coverage on the reference sequence.

11.  Before publication, submit raw NGS data to Sequence Read Archive (SRA) and give accession numbers in the text.

12.  Line 105 and 170 – "Three paralogous gene pairs including RrOSC1 and 105 RrOSC3, RrOSC2 and RrOSC10, and RrOSC5 and RrOSC18 derived by tandem replications according to the synteny analysis" There is no evidence in the results to confirm the formation of these pairs of genes by tandem duplication. Figure 2 shows that these genes differ dramatically in the nucleotide sequence length, the intron/exon arrangement, and the structure of conserved motifs.

13.  Line 66 – “involved in the terpenoids (synthesis?) of R. rugosa haven't been reported”

14.  Line 67 – “systematic analysis of predicted OSC (gene?) family of R. rugosa (RrOSC)”

15.  Line 71 - unnecessary period at the end of a line of text.

Author Response

The manuscript " Systematic Identification and Analysis of OSC Gene Family of Rosa rugosa Thunb."  in my opinion, presents interesting and valuable data. It fits in well with the scope of the journal. However, the manuscript requires major revisions and supplementing important information. Particularly the Materials and Methods are not sufficiently described, and this section needs to be completed.

Response: We appreciate the reviewer’s the overall positive comments and suggestions and we have totally revised the M&M.

  1. Figures 1. and 2. are small and unreadable. Figure 2. should be enlarged and possibly divided. Instead, you can remove Figure 6, which in my opinion, is not necessary (or move it to the supplement)

Response 1: Thank you for this point. We have provided images with higher pixel.

  1. The authors several times n the article indicated they identified 18 OSC genes, but the RrOSC4 gene is missing in table S1 and in all analyses. All information about this gene is missing except that it is short and possibly a pseudogene ( line 72). Size and chromosomal location have not been estimated. How was this gene identified, and why was it not included in the analysis? Was it observed in the transcriptome?

Response 2: Thank you for this point. The OSC family is extremely conservative, and usually contains two conservative domains: the squalene-hopene cyclase N-terminal domain and the squalene-hopene cyclase C-terminal domain. The predicted RrOSC4 protein is 142 AA in length which is shorter than average length. The short RrOSC4 contains only squalene-hopene cyclase N-terminal domain while lacks the C-terminal domain. In roots, stems and flowers, abundance of RrOSC4 is too low(FPKM < 0.3)to distinguish from the RNA-seq noise (e.g., clip mapping reads). Considering that it could be a pseudogene or a function unknown protein, we did not include it in the study. Anyway, we have added the size and chromosomal location of this supposed OSC gene in the Table S1.

  1. What did Gene IDs mean in Table S1? Are the sequences of RsOSC genes accessible in any databases? The sequences of all new RrOSC genes should be submitted to the database, and the accession numbers should be enclosed in the article.

Response 3: IDs in Table S1 are ID in genome of R. rugosa ‘Zi Zhi’ which is sequenced by PacBio RS II and assembled by our lab (unpublished, not accessible now). We have supplied the RrOSC protein sequences in Table S1.

  1. Complete table S1 with additional information: length of the nucleotide sequence, number of exons, and location on the chromosome.

Response 4: Thank you for this point. We have completed it.

  1. Line 211. Identification of RrOSCs –description is insufficient. Based on the text, I can't understand how the authors obtained the RrOSCAs amino acid sequences. No information about reference sequences was used to predict the OSCs sequences.

Response 5: Thank you for this point. The genome of R. rugosa “Zi Zhi used in this study (genome sequences and GFF files) could not be public resources until further the paper of genome would be published. We have supplied the mRNA and DNA sequences of Rr OSCs in Table S3 and S4.

  1. No information about the proteome of R. rugosa "Zi Zhi." Is it accessed in any database? Where is the proteome come from? What tissue was this transcriptome prepared?

Response 6: Thank you for this point. The proteome of R. rugosa "Zi Zhi." was the CDS translation of R. rugosa ‘Zi Zhi’ genome (according to GFF annotation). As part of the genome data of R. rugosa “Zi Zhi”, it has not been published yet. Anyway, we have supplied the RrOSC protein sequences in Table S1. The tissues of transcriptome are listed in 4.1 and Figure 6.

  1. The authors poorly describe the process of gene identification and their location in the genome. For example, what bioinformatics tools were used for genome localization? What was the reference genome?

Response 7: Thank you for this point. We have increased the process description of gene identification and their location in the genome.

  1. Line 216. The subcellular localization –  where is the result of the subcellular localization analysis described in the article?

Response 8: Thank you for this point. We think the result of the subcellular localization is un important, we have deleted it.

  1. Phylogenetic analyses - lack basic information regarding phylogenetic analysis:  alignment methods and program settings. What substitution model was used in the phylogenetic tree construction, and how was this model established? A table containing the percentage of gene similarity at the AA level would be helpful. It could be added to the supplement.

Response 9: We have added the basic information in M&M. The alignment methods MAFFT was used for sequence alignment. Different models/parameters have been tried for NJ-tree building by of MEGA X and we got similar topological structures. It indicated that NJ method was insensitive to the substitution model. Indeed, we used the p-distance model as the substitution model and uniform rates as rates among sites. The AA similarity matrix was followed (it is not closely related to our study so we did not attach it to the additional table).

  1. Line 231 - Information about NGS sequencing results should be provided: sequencing quality - Q30 value, percentage of rejected reads after quality trimming, and alignment coverage on the reference sequence.

Response 10: We have added Q30 value, percentage of clean reads and reference genome coverage to Table S7.

  1. Before publication, submit raw NGS data to Sequence Read Archive (SRA) and give accession numbers in the text.

Response 11: We have submitted all RNA-seq raw data to SRA of NCBI (PRJNA725330) and BIG Sub of CNBC (China National Center for Bioinformation, PRJCA012932).

  1. Line 105 and 170 – "Three paralogous gene pairs including RrOSC1 and 105 RrOSC3, RrOSC2 and RrOSC10, and RrOSC5 and RrOSC18 derived by tandem replications according to the synteny analysis" There is no evidence in the results to confirm the formation of these pairs of genes by tandem duplication. Figure 2 shows that these genes differ dramatically in the nucleotide sequence length, the intron/exon arrangement, and the structure of conserved motifs.

Response 12:  We have checked the tandem duplication genes and revised the misused gene pairs as RrOSC2-RrOSC3, RrOSC8-RrOSC9, RrOSC14-RrOSC15. The ‘duplicate_gene_classifier’ program (https://github.com/wyp1125/MCScanX) of MCScanX (Wang Y. Nucleic Acids Res. 2012 Apr;40(7):e49) would predicted the gene duplication events (including 0 singleton, 1 dispersed, 2 proximal, 3 tandem and 4 WGD/segmental). We have supplied the output data of gene duplication analysis of R.rugosa genome as Table S3 and S4.

  1. Line 66 – “involved in the terpenoids (synthesis?) of R. rugosa haven't been reported”

Response 13: Thank you for this point. We have revised this sentence. In 2005, rosamultin (a triterpene saponin) was reported that it was extracted from R. rugosa. Our lab also detected rosamultin and its upstream substance ursolic acid in R. rugosa. Therefore, we believe that there is a triterpene synthesis pathway in R. rugosa.

  1. Line 67 – “systematic analysis of predicted OSC (gene?) family of R. rugosa (RrOSC)”

Response 14: Thank you for this point. We have revised line 67 to make it clear.

  1. Line 71 - unnecessary period at the end of a line of text.

Response 15: Thank you for this point. We have modified it.

Reviewer 2 Report

The manuscript is accepted after revision in abstract, Introduction should be improved with latest references, Discussion part should also be improved through justification research findings with other researchers.

Author Response

The manuscript is accepted after revision in abstract, Introduction should be improved with latest references, Discussion part should also be improved through justification research findings with other researchers.

Response : We appreciate the reviewer’s the overall positive comments and suggestions and we have revised the introduction and discussion and improved with latest references.

Reviewer 3 Report

This manuscript provides a detailed analysis of the oxidosqualene cyclase family in Rosa rugose. This gene family is involved in the first step of triterpenoids biosynthesis, being triterpenoids compounds with proven pharmacological activities. The study includes a phylogenetic and motifs analysis of all OSC proteins. Strong points of this work are that it includes an expression analysis of OSC genes in different tissues and the functional characterization in yeast of one of the identified genes. The text is simple and clear and it reads well to me. A few minor modifications and additions can help to improve the manuscript:

I can see the potential of RrOSC genes for metabolic engineering in other plant species or other organisms via heterologous overexpression. For example, plants of the Nicotiana genus are commonly used as biofactories for metabolite production. I think that previous studies using OSC and other key triterpenoid biosynthetic genes in metabolic engineering could be cited in the introduction/discussion. It would add more significance to the work described here.

In the phylogenetic tree, OSCs from different plant species are included. I think that all species included in the tree should be mentioned either in the results or in the M&M section. Are there beta-amyrin synthases in related species? Why do you think they are not present in Rosa?

Line 87. It is not clear what the authors mean by “located by 17-19 introns”. Encoded by x exons?

Fig2A and 2C show that all motifs are highly conserved. Can the authors mention if any of the motifs has a known function?

Can the authors suggest an explanation for the big differences in expression between open-air and tissue culture roots. Why the difference is so big and not always in the same direction? For example, OSC12 has higher expression in tissue culture roots and the expression of OSC18 is higher in open-air roots. Moreover, the authors state that the expression of OSC10, 12, 13, and 18 is high in roots. This is true only for open-air roots.

Author Response

This manuscript provides a detailed analysis of the oxidosqualene cyclase family in Rosa rugose. This gene family is involved in the first step of triterpenoids biosynthesis, being triterpenoids compounds with proven pharmacological activities. The study includes a phylogenetic and motifs analysis of all OSC proteins. Strong points of this work are that it includes an expression analysis of OSC genes in different tissues and the functional characterization in yeast of one of the identified genes. The text is simple and clear and it reads well to me. A few minor modifications and additions can help to improve the manuscript:

Response: We appreciate the reviewer’s the overall positive comments and suggestions

I can see the potential of RrOSC genes for metabolic engineering in other plant species or other organisms via heterologous overexpression. For example, plants of the Nicotiana genus are commonly used as biofactories for metabolite production. I think that previous studies using OSC and other key triterpenoid biosynthetic genes in metabolic engineering could be cited in the introduction/discussion. It would add more significance to the work described here.

Response 1: Thank you for this nice point. We have increased the discussion on key triterpene biosynthesis genes in metabolic engineering.

In the phylogenetic tree, OSCs from different plant species are included. I think that all species included in the tree should be mentioned either in the results or in the M&M section. Are there beta-amyrin synthases in related species? Why do you think they are not present in Rosa?

Response 2: Thank you for this point. Several representative genes are mentioned in the discussion section to discuss the function of R. rugosa genes. Though β-Amyrin synthases are present in many plant species. Obviously, there is no R. rugosa OSC member in the β-amyrin synthases group, but it cannot be ruled out that it appears in the form of mixed products. We suspected that RrOSC/RrOSCs in the diverse products groups (GI) would generate β-amyrin as main component of its mix products.

Line 87. It is not clear what the authors mean by “located by 17-19 introns”. Encoded by x exons?

Response 3: Thank you for this question. We have revised this as “located by 18-20 exons”.

Fig2A and 2C show that all motifs are highly conserved. Can the authors mention if any of the motifs has a known function?

 Response 4: Thank you for this question. We have searched motifs of OSCs to detect motifs involved in the subcellular localization or catalytic activity, but no functional motifs have been identified in these motifs.

Can the authors suggest an explanation for the big differences in expression between open-air and tissue culture roots. Why the difference is so big and not always in the same direction? For example, OSC12 has higher expression in tissue culture roots and the expression of OSC18 is higher in open-air roots. Moreover, the authors state that the expression of OSC10, 12, 13, and 18 is high in roots. This is true only for open-air roots.

Response 5: Thank you for this question. The open-air roots were sampled from 3-year-old seedings of R.sugosa ‘Zizhi’ and the tissue culture roots were sampled from microcutting seedings of R.sugosa ‘Zizhi’ (2-month-old). The open-air roots and tissue culture roots derived from adventitious roots, but the tissue culture roots were quite younger (white color, non- lignification) than open-air roots (brownness, semi-lignification). It is no strange that one OSC expressed lower or higher between the two kinds of roots since their quite different physiological status or developmental stage. For example, RrOSC12 may played more important roles in younger roots than in older roots.

Reviewer 4 Report

#1:  Gene name are not consistent throughout the paper. In some places, gene names are italic with block letters whereas in other places they are written as small letters with no italic form. Wild-type gene name should be italic with block letters and protein name should be with block letters.

#2: Method section are poorly presented. Particularly, 4.3, 4.4, and 4.5

#3 line 28-29: Please rewrite

#4 line 31: biochemical reaction, 2,3-Oxidosqualene was is the common

#5 line 32-33: Rewrite

#6 line 53: Provide reference

#7 71: Rewrite

#8 75" Rewrite

#9: please provide high resolution image for all figures. 

#9 86-88, 103, 158: Rewrite

Author Response

Gene name are not consistent throughout the paper. In some places, gene names are italic with block letters whereas in other places they are written as small letters with no italic form. Wild-type gene name should be italic with block letters and protein name should be with block letters.

Response: We appreciate the reviewer’s comments and suggestions and we have revised the gene names.

#2: Method section are poorly presented. Particularly, 4.3, 4.4, and 4.5

Response 1: We have revised these sections.

#3 line 28-29: Please rewrite

Response 2: We have revised this sentence.

#4 line 31: biochemical reaction, 2,3-Oxidosqualene was is the common

Response 3: We have revised this verb form.

#5 line 32-33: Rewrite

Response 4: We have revised this sentence.

#6 line 53: Provide reference

Response 5: We have provided the reference [13] of the reference list.

#7 71: Rewrite

Response 6: We have revised this sentence.

#8 75" Rewrite

Response 7: We have revised this confusing sentence.

#9: please provide high resolution image for all figures. 

Response 8: We have provided high resolution image (vectorgraph in PDF).

#9 86-88, 103, 158: Rewrite

Response 9: We have revised the 3 sentences.

Round 2

Reviewer 1 Report

The authors corrected and introduced the necessary changes to the manuscript. Therefore, the article can be published in this form. However, I am still doubtful because the results presented in the article are based on the unpublished genome of R. rugosa 'Zi Zhi'. In my opinion, genome data should be made accessed before publication.

Author Response

The authors corrected and introduced the necessary changes to the manuscript. Therefore, the article can be published in this form. However, I am still doubtful because the results presented in the article are based on the unpublished genome of R. rugosa 'Zi Zhi'. In my opinion, genome data should be made accessed before publication.

Answer: We regret that we can’t submit the genome data to the open access database at present. We are drafting the manuscript of R. rugosa 'Zi Zhi' genome including genome-sequencing, assembly of chromosomes and resequencing for quantitative trait locus identification. All these raw data would be released along with publication of the manuscript.

Anyway, all the authors agree to share the genome sequence file (.fasta) and annotation file (.gff3) in a limited range by written request. We have seed this information to editor. If any researcher requests the genome information, he can contact with the email of corresponding author ([email protected]).

Reviewer 4 Report

Please, check the use of English throughout the manuscript

Author Response

Please, check the use of English throughout the manuscript

Answer: We have revised the language throughout the manuscript.